# Amyloid Formation in Nanoliter Droplets

**DOI:** 10.3390/ijms23105480

**Published:** 2022-05-13

**Authors:** Da Yeon Cheong, Wonseok Lee, Insu Park, Jinsung Park, Gyudo Lee

**Affiliations:** 1Department of Biotechnology and Bioinformatics, Korea University, Sejong 30019, Korea; 2017270450@korea.ac.kr; 2Interdisciplinary Graduate Program for Artificial Intelligence Smart Convergence Technology, Korea University, Sejong 30019, Korea; 3Department of Electrical Engineering, Korea National University of Transportation, Chungju 27469, Korea; wslee@ut.ac.kr; 4Nick Holonyak Jr. Micro and Nanotechnology Laboratory, University of Illinois at Urbana-Champaign, Urbana, IL 61801, USA; insupark@illinois.edu; 5Department of Biomechatronic Engineering, College of Biotechnology and Bioengineering, Sungkyunkwan University, Suwon 16419, Korea

**Keywords:** lysozyme, amyloid formation, nanoliter droplet, fluorescence assay, atomic force microscopy, polymorphism

## Abstract

Processes that monitor the nucleation of amyloids and characterize the formation of amyloid fibrils are vital to medicine and pharmacology. In this study, we observe the nucleation and formation of lysozyme amyloid fibrils using a facile microfluidic system to generate nanoliter droplets that can control the flow rate and movement of monomer-in-oil emulsion droplets in a T-junction microchannel. Using a fluorescence assay, we monitor the nucleation and growth process of amyloids based on the volume of droplets. Using the microfluidic system, we demonstrate that the lag phase, which is vital to amyloid nucleation and growth, is reduced at a lower droplet volume. Furthermore, we report a peculiar phenomenon of high amyloid formation at the edge of a bullet-shaped droplet, which is likely due to the high local monomer concentration. Moreover, we discovered that amyloid fibrils synthesized in the nanoliter droplets are shorter and thicker than fibrils synthesized from a bulk solution via the conventional heating method. Herein, a facile procedure to observe and characterize the nucleation and growth of amyloid fibrils using nanoliter droplets is presented, which is beneficial for investigating new features of amyloid fibril formation as an unconventional synthetic method for amyloid fibrils.

## 1. Introduction

The transformation of soluble native proteins into insoluble aggregates is the general characteristic of amyloid formation, which can induce neurodegenerative diseases such as chronic traumatic encephalopathy, Parkinson’s disease, and Alzheimer’s diseases [1,2,3,4,5]. In the past decade, many methodologies (e.g., heating [6], microwave irradiation [7,8], and thermal cycling [9]) have been investigated and developed for monitoring the nucleation, rapid growth, secondary nucleation, and growth of amyloids to elucidate the origins of neurodegenerative disorders.

Among them, microdroplets, as microchemical reactors, have been used to observe the spatial propagation of amyloid assembly, monitor the nucleation of amyloids, and characterize the formation of amyloid fibrils [10]. In particular, the presence of amyloid proteins stained with fluorescent molecules (e.g., thioflavin-T (ThT)) at the oil–water interface was investigated through droplet and microfluidic systems. Microfluidic systems have been developed as an alternative strategy to study the formation of amyloids, which can overcome the limitation of bulk solution methods [11]. For example, by regulating the flow rate of a water or oil solution, the volume of an amyloid solution can be controlled in a microfluidic system by modifying the aspect ratio of the droplet [12]. Moreover, this enables the accurate modeling of amyloid formation and the quantitative characterization of fibrillation [10]. Despite these efforts, the elucidation of the mechanisms by which native proteins are amyloidized in nanoliter droplets remains challenging.

In this study, we monitored the fibrillation of hen egg-white lysozyme (HEWL) using a facile microfluidic system that can control the flow rate, volume of droplets, and movement of water-in-oil emulsions for synthesizing nanoliter droplets. Specifically, we custom developed a microfluidic system with a T-junction microchannel and investigated the formation of HEWL aggregates for 76 h by monitoring the nanoliter droplets. By regulating the flow rate of oil and water containing HEWL monomers, we successfully controlled the volumes from the micro to nanoliters. In the droplet system, we monitored the nucleation and growth processes of amyloid using a ThT fluorescence assay [13]. Moreover, we used atomic force microscopy (AFM) to compare the structural information of amyloid fibrils synthesized by the general synthetic method and the microfluidic system [6].

## 2. Results

### 2.1. Microfluidic System Generating Nanoliter Droplets for Observation of Amyloid Formation

To construct a microfluidic system that can generate nanoliter droplets, we assembled a tee connector and tubing with a sufficiently narrow inner diameter (ID) (i.e., 0.5 mm). The structure of the fabricated microfluidic system and the experimental setup are shown in Figure 1.

The system comprised two inlets, one for water injection and another for oil injection. Hundreds of nanoliter droplets were generated, which caused the oil flow to cut the water flow. The generated droplets were stored in the tubing connected to the outlet, and the stored droplets were collected by cutting and hermetically sealing the tubing. The droplets collected were heated in an oven for amyloid fibril synthesis (see the Section 5). Heat promotes amyloid nucleation by partially unfolding the protein into a transition state that favors the formation of amyloid fibrils [14]. In addition, heat also can accelerate the Brownian motion of protein molecules, causing more intermolecular interactions [15]. Including these processes, we established a platform based on a droplet generation system to synthesize amyloid fibrils in a relatively small and tight space.

### 2.2. Size Modulation of Nanoliter Droplets in Microfluidic System

To confirm the performance of the microfluidic droplet generator, we controlled the droplet size. By controlling the ratio of the flow rates of water and oil in the microfluidic system, the length and volume of the generated droplets can be controlled. In the experiment, the oil flow rate was fixed at 20 µL/min, and the droplet size was adjusted by varying the water flow rate (20–80 µL/min; Figure 2a–d). As the ratio of the water flow rate increased from 1:1 to 1:4, the droplet size increased continuously from 100 to 300 nL (Figure 2e). To more clearly observe the correlation between the Inlet 1 and 2 flow rate ratio and droplet size, we initially set the flow rate to 1:4 and then gradually decreased the flow rate of Inlet 2; subsequently, we observed the size of the generated droplet. Our observation confirmed that as the flow rate of Inlet 2 (to which the protein solution was supplied) decreased, the size of the formed droplets decreased continuously. Figure 2f shows the change in the droplet size as the flow rate of Inlet 2 decreased. Based on the result, it was confirmed that the size of the droplets generated by the microfluidic system can be easily controlled.

### 2.3. Fluorescence Assay for Observation of Amyloid Formation in Nanoliter Droplets

ThT fluorescence analysis was performed to observe amyloid formation inside the droplets. ThT is a fluorescent substance that quantifies the cross β-sheet structure of amyloid fibrils, and the deposition of amyloid fibrils can be observed based on the ThT fluorescence intensity [16]. In the experiment, droplets of five different volumes were collected using the established droplet generation system, and the change in ThT fluorescence intensity as a function of heating time was analyzed (Figure 3a–e). The intensity recorded over time was fitted by sigmoidal curves (Figure 3f). We calculated the lag times, and it was discovered that the smaller the droplet volume, the shorter the lag time (Figure 3g).

The change in the lag time vs. the droplet volume showed an exponential growth curve, which is contrary to the results of previous studies. Knowles et al. observed the primary nucleation and spatial propagation of amyloids inside small pL-scale droplets, where it was discovered that the smaller the droplet volume, the longer the lag time [10]. For extremely small droplets containing only a few protein molecules, the lag time is long because nucleation events rarely occur. In the abovementioned study, the aspect ratio was increased significantly by actuating the droplet into an extremely narrow channel to capture a single nucleation site. Because a droplet with a large aspect ratio is not suitable for convection, the movement of its protein molecules may be inhibited. The nanodroplet investigated in this study appeared to be affected by variables other than the number of nucleation events occurring in the system. It is presumed that a smaller vesicle system results in a higher thermal conductivity and convection degree, and hence a higher nucleation rate because the unfolded peptide chains interact with each other at a higher frequency. For example, Ferreira et al. observed that the current inside and outside a droplet changes with the droplet size [17], whereas Maeki et al. reported that the number of crystals formed by protein molecules changes with the droplet size [18]. These studies support the hypothesis regarding the correlation among system volume, convection, and nucleation rate mentioned above, which must be elucidated for the rapid and efficient production of amyloid fibrils. In addition, the significantly different fibrillation lag time between the insulin in Knowles’s research (100–200 min) [10] and the HEWL in this study (14–63 h) is attributable to the relatively large molecular weight of HEWL. Proteins with large molecular weights do not readily nucleate because they cannot easily attain the transition state owing to their complexly folded long polypeptide chains [19].

### 2.4. Rapid Amyloid Nucleation in Asymmetric Droplet

In this study, we observed an interesting phenomenon. Occasionally, tiny gaps were formed from the sealed region of the collected tubing over time during incubation at 60 °C, which was attributed to evaporation through the gap. Then, a biased pressure was applied to the droplet, which induced its deformation. An asymmetrically deformed droplet resembling a bullet with a concave meniscus and a sharp edge was created at the section that was in contact with the inner side of the tubing (Figure 4a).

By this deformation, a surface tension gradient can be formed on the droplet surface. This gradient induces the Marangoni effect, whereby unfolded peptide chains are locally deposited at the edge and form a relatively high local concentration [20,21]. Owing to the significant molecular crowding at the corner, rapid nucleation was observed, as shown by the results of bright-field and fluorescence microscopies (Figure 4b,c, respectively). To examine the effect of protein concentration on the lag time, an experiment with a lower protein concentration was additionally conducted. The lag time for 1 wt% HEWL droplets was 62.8 h, which was 1.6-fold longer than that of 3 wt% HEWL droplets (39.9 h) (Appendix A). This result supports that the high local concentration could trigger rapid amyloid nucleation. A cross-sectional profile analysis of the fluorescence images confirmed the appearance of a high ThT intensity at the edge of the droplet (Figure 4d). This phenomenon suggests that amyloid deposition may occur more easily in cells that form sharp edges in the body (e.g., neurons and smooth muscle cells) [22,23]. Furthermore, it may be associated with the predominant occurrence of amyloid deposition at interstitial sites in cells [24].

### 2.5. Polymorphism of HEWL Amyloid Fibrils Synthesized within Nanoliter Droplet

The surrounding environment is vital to amyloid formation [25]. The morphology of amyloid fibrils can change with the environmental conditions; consequently, amyloid fibrils may exhibit physical properties that are different from those of original fibrils, which affect their pathogenicity—this is known as amyloid polymorphism [19,26]. Therefore, the occurrence of amyloid polymorphisms in nanoliter droplets should be verified. First, it was speculated that the morphology of amyloid fibrils fibrillated in droplets contained in a miniature system would differ from that of amyloid fibrils synthesized in a large system in a bulk solution. To support our hypothesis, we performed an AFM imaging of amyloid samples extracted from the droplets and compared them with HEWL amyloid fibrils synthesized via conventional bulk heating (Figure 5a,d).

Based on the AFM images, we performed a statistical analysis of the amyloid fibrils with respect to the contour length and fibril diameter [6]. The average contour length of amyloid fibrils synthesized in a bulk solution was 230 ± 104 nm, but fibrils measuring up to 2 μm in length were observed as well (Figure 5b) [27]. By contrast, the contour length of fibrils synthesized inside the droplet was 137 ± 78 nm (Figure 5e). We confirmed that there was a significant difference between the lengths of amyloid fibril produced in bulk solution and produced in nanodroplets (*p*-value < 0.0001). Moreover, the diameter of amyloid fibrils synthesized in the bulk solution was 2.45 ± 0.95 nm (Figure 5c), whereas the diameter of fibrils synthesized inside the droplets was 3.98 ± 1.32 nm (Figure 5f). We also confirmed a significant difference between the diameters of fibril produced in bulk solution and produced in nanodroplets (*p*-value < 0.0001). In general, the length of the amyloid fibril synthesized inside the droplet was 79.2% shorter and 62.4% thicker than that of the amyloid fibrils synthesized from the bulk solution. These structural differences (i.e., polymorphism) are attributable to the narrow space of the droplet, as compared with the bulk solution [28]. Owing to the relatively tight space of the nanodroplets, the amyloid fibrils grew radially and did not elongate in the axial direction [29]. A thicker fibril diameter implies an active secondary nucleation on the amyloid surface [30].

## 3. Discussion

In droplet-based amyloid fibril formation, droplet size and deformed droplet edge were discussed as two factors affecting lag time. The smaller the droplet size, the stronger the degree of convection, causing more frequent interactions between protein molecules. When an edge is formed by asymmetric pressure on the droplet, convection occurs due to the Marangoni effect, and protein molecules are concentrated on the edge to form a high local concentration. Briefly, droplet size and edge affect the convection and protein concentration and change the lag time. This suggests that the shape and size of the system must be considered when studying amyloid nucleation.

We also revealed that HEWL amyloid fibrils synthesized using nanoliter droplets had a shorter length and thicker diameter than fibrils synthesized from a bulk solution. This finding implies that amyloids produced in a tight space in the body will exhibit significantly different properties from those produced in a bulk solution [31]. To date, studies of amyloid fibril polymorphisms have primarily focused on the characteristics of solutions that constitute the surrounding environment, such as the pH, ion strength, additives, and stirring rate, but not the system volume [25,32,33,34,35,36]. Considering the results of this study, it was clear that not only the characteristics of the solution, but also the shape and size of the space affect the behavior of amyloid molecules. Because the region where amyloid fibrils are likely to be formed in our body (e.g., cells, extracellular matrix, and blood vessels) is near an extremely small space instead of a large space [37,38,39], amyloid fibrils formed in micro- or nanosizes other than those from a bulk solution should be investigated. Hence, we believe that the facile microfluidic system for generating nanoliter droplets presented herein provides an advanced strategy for a comprehensive understanding of the conversion of native proteins into amyloids in a small space. Moreover, this study can provide a basis for investigating the difference in the morphology of amyloids formed in vitro and in vivo [40,41]. Microfluidic technology that can create and visualize a microenvironment is essential for investigating the polymorphism of amyloids based on the system volume. In the future, if amyloid fibrils with polymorphism are synthesized using microfluidic droplet generation systems and cytotoxicity experiments are conducted on these amyloids, then the specific physical environment that poses a pathological threat can be determined while excluding risk factors.

Our approach can be applied to manufacture a high-throughput drug screening system based on nanoliter amyloid droplet arrays. The droplets made with our system can be converted into solid amyloid particles by using evaporation (Appendix A) and used for applications such as drug screening. The amyloid-degrading ability of the drug can be determined by treating the amyloid particles and observing whether they are decomposed. In addition, when the drug is added to the protein solution in advance and it has been observed through fluorescence microscopy whether amyloid fibrils are formed in the generated droplet, it is possible to determine the amyloid inhibition ability of the drug. Our simple system allows droplet-based experiments without expensive equipment, which is expected to accelerate the drug discovery.

## 4. Conclusions

In this study, we utilized T-junction microfluidic channels to generate nanoliter droplets containing HEWL monomeric proteins and observed the nucleation and fibril formation of HEWL amyloids. By analyzing the change in ThT fluorescence intensity of the droplet over time, we confirmed that the nucleation and growth process of amyloids varied based on the droplet volume. Based on the used assay, we demonstrated that the lag phase reduced as the droplet volume decreased, owing to frequent collisions between HEWL monomers. Moreover, we discovered that the droplets deformed from a spherical shape to a bullet shape with a concave meniscus owing to their asymmetric evaporation. In the vicinity of a concave meniscus, rapid amyloid nucleation occurred, which is attributed to the condensation of aggregates to a high local concentration of monomers. This implies that further studies of amyloid fibrillation based on multiform systems are needed. In particular, the polymorphism of amyloid fibril according to the system volume (bulk vs. nanodroplets) shown in this study suggests the possibility of amyloid polymorphism depending on the dimension of the system. This study provides a new perspective on the mechanism of amyloid fibril formation, which will lead to a deeper understanding of amyloid formation that occurs in a narrow and complicated space in the human body.

## 5. Materials and Methods

### 5.1. Reagents and Materials

HEWL, hydrochloric acid (HCl), thioflavin-T (ThT), octanol, and 2-propanol were purchased from Sigma–Aldrich (St. Louis, MO, USA). Distilled water from Gibco was purchased from Thermo Fisher Scientific (Waltham, MA, USA). A Whatman™ 0.2 µm polyvinylidene difluoride (PVDF) membrane syringe filter (GH Healthcare, Chicago. Illinois, USA) was used for to filter the solutions. A VHP tee connector and perfluoroalkoxy alkane (PFA) tubing were purchased from IDEX (Northbrook, IL, USA). Water repellents (Aquapel Glass Treatment, PGW Auto Glass, LLC, Cranberry Township, PA, USA) were purchased from commercial sources.

### 5.2. Preparation of HEWL Solution and HEWL Amyloid Fibrillation in Bulk Solution

A 1 wt% and 3 wt% HEWL solution was prepared by dissolving HEWL in pH 2 distilled water, which was titrated with 1 M HCl [42]. The ThT concentration in the HEWL solution was adjusted to approximately 20 µM by adding 1 mM ThT to the solution [16]. The prepared solution was filtered through a 0.2 µm syringe filter to eliminate large aggregates. To form HEWL amyloid fibrils in the bulk solution, the HEWL solution was filtered through a 0.2 µm syringe filter. Subsequently, 3 mL of the solution in a 6 mL vial was incubated for 5 d in a 60 °C oil bath.

### 5.3. Generation of Nanoliter Droplets

A nanoliter droplet generation system was constructed using a VHP tee connector and PFA tubing, and their ID was unified to 0.020” (or 0.50 mm). All channels were filled with a water repellent and then incubated for 5 min to achieve a hydrophobic coating. Subsequently, they were rinsed with 2-propanol and dried with N_2_ gas. Syringe pumps (NEXT ZS100, Next Pump, Gwangmyeong, Korea) were used to control the fluid flow of the nanoliter-droplet-generation system. The as-prepared HEWL solution containing ThT was used as the water flow, and octanol was used as the oil flow. The flow rates used in this study were 20 and 20–80 µL/min for the oil and water flows, respectively. The generated droplets were observed using a bright-field microscope (Nikon Eclipse Ni, Nikon, Tokyo, Japan), and the droplet length was measured using an imaging software (NIS-Elements BR, Nikon, Tokyo, Japan). Thirty droplets were analyzed for each condition for length and volume measurements. An unpaired t-test was performed to identify significant differences for each condition.

### 5.4. Fluorescence Assay for Detection of Amyloid Fibrillation in Nanoliter Droplets

The tubing in which the nanoliter droplets were captured was obtained via cutting, and both ends of the opened tubing were sealed firmly to prevent evaporation. Metal pins were inserted at both ends of the tubing, and then the joint was thoroughly wrapped with Teflon tape to avoid evaporation. To form amyloid fibrils, the tubes obtained were incubated at 60 °C for 5 d. The ThT fluorescence deposition of the droplets was imaged using a fluorescence microscope (Nikon Eclipse Ni, Nikon, Tokyo, Japan). Illumination was supplied by a precentered fiber illuminator (Intensilight C-HGFI, Nikon, Tokyo, Japan), and a filter cube was used to detect ThT fluorescence (450 nm excitation and 490 nm emission) [16]. The ThT fluorescence intensity for each droplet was measured using imaging software (NIS-Elements BR, Nikon, Tokyo, Japan). In this experiment, three droplets were analyzed for each condition. Sigmoidal fitting, which follows Hill function, was performed to calculate the lag time. The lag time was defined as the required incubation time for half-maximal fluorescence intensity (i.e., the time point at max/2).

### 5.5. AFM Imaging for Structural Analysis of HEWL Amyloid Fibrils

The fibril morphology was investigated using a MultiMode VIII atomic force microscope (Bruker, Billerica, Massachusetts, USA) operating in the tapping mode. To prepare samples of amyloid fibrils formed in the bulk solution for AFM, the fibrils were diluted with distilled water (pH 2) to a concentration of 0.1 wt%, and 20 µL aliquots of the solution were deposited onto freshly cleaved mica, incubated for 5 min, rinsed with 300 µL of distilled water, and dried with N_2_ gas [6]. For amyloid fibrils formed in nanoliter droplets, the incubated nanoliter droplets were collected and centrifuged (13,000 rpm, 15 min) to separate the aqueous and oil phases. Subsequently, the supernatant containing oil was discarded, and 20 µL of the aqueous solution was deposited onto the mica substrate and subjected to the same procedure as the fibril formed in the bulk solution.

### 5.6. Statistical Analysis of AFM Images

A morphology analysis of the fibrils was performed using Nanoscope Analysis software (version 2.0). To obtain the fibril diameter distribution, 200 fibrils were analyzed for each sample. Their diameters were determined by cross-sectioning and measuring for the maximum height of the fibrils. The contour length of 100 fibrils for each sample was analyzed based on the AFM images using the Easyworm software [43]. An unpaired *t*-test was performed to confirm a significant difference between the conditions for each acquired diameter and length data of fibrils.

## Figures and Tables

**Figure 1 ijms-23-05480-f001:**
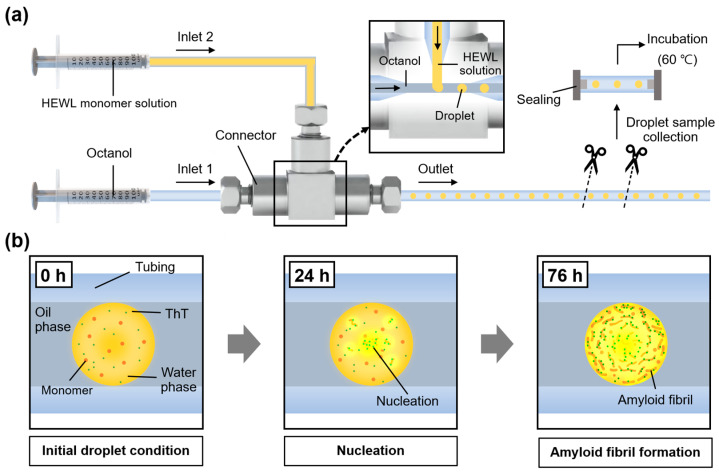
Schematic illustration of nanoliter-droplet-generation model for investigating amyloid formation. (**a**) Design of microfluidic system for droplet generation. (**b**) Illustration of time-resolved observation inside nanoliter droplet from nucleation to fibrillation of HEWL over time (0–76 h).

**Figure 2 ijms-23-05480-f002:**
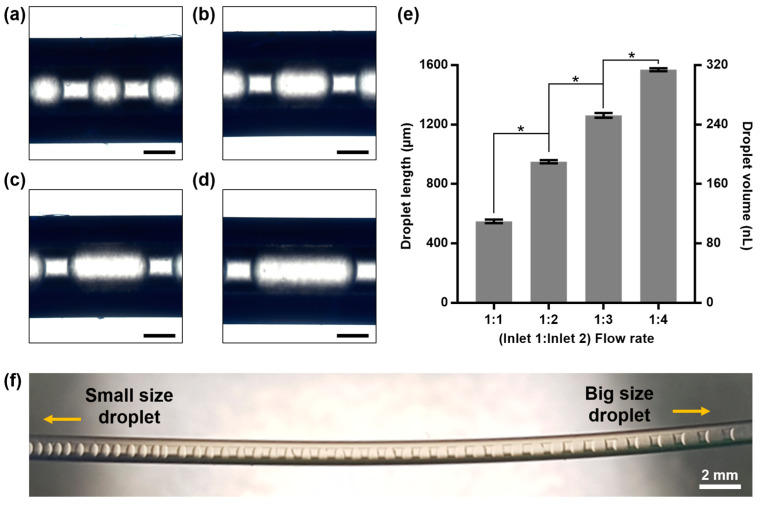
Volume control of nanoliter droplets in microfluidic system. Bright-field images of droplets generated under different flow rate ratios of oil (from Inlet 1) and protein solution (from Inlet 2) ((**a**) 1:1; (**b**) 1:2; (**c**) 1:3; (**d**) 1:4). Flow rate of oil was 20 μL/min, and flow rate of protein solution was varied from 20 to 80 μL/min. Scale bar is 500 μm. (**e**) Droplet lengths and volumes based on different ratios of flow rate between oil and protein solution (* *p*-value < 0.0001). (**f**) Size control of droplets by gradually altering flow rate of the protein solution.

**Figure 3 ijms-23-05480-f003:**
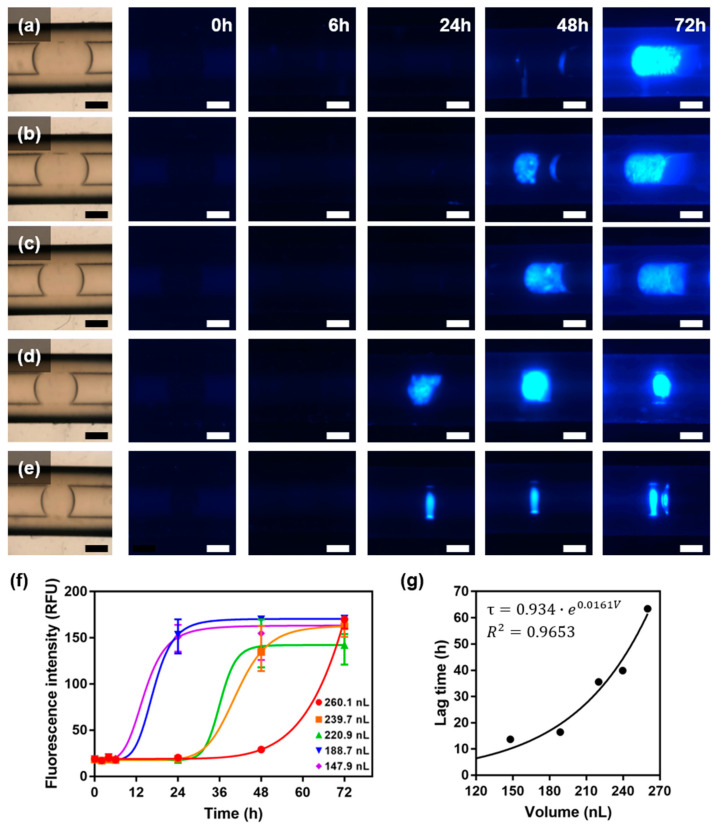
Amyloid fibrillation in nanoliter droplets. Bright-field images of nanoliter droplets based on volume ((**a**) 260.1 nL; (**b**) 239.7 nL; (**c**) 220.9 nL; (**d**) 188.7 nL; (**e**) 147.9 nL), and corresponding fluorescence images of nanoliter droplets during incubation. Scale bar is 500 μm. (**f**) ThT fluorescence assay for detection of fibrillation. (**g**) Lag times (τ) of amyloid fibrillation in different volumes (V) of droplets.

**Figure 4 ijms-23-05480-f004:**
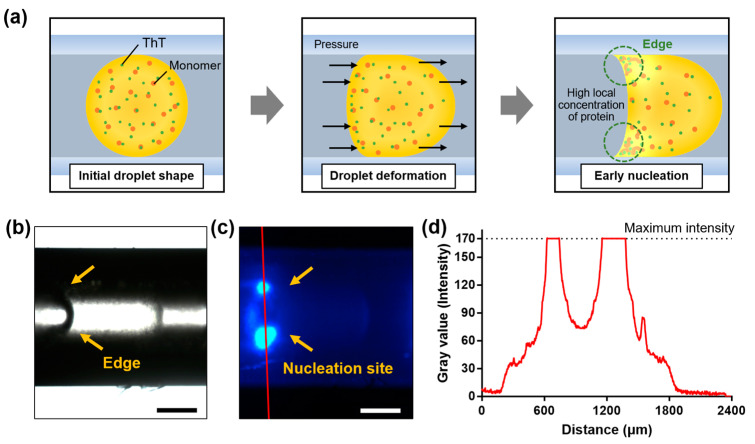
Rapid amyloid nucleation at vicinity of concave meniscus of droplets. (**a**) Schematic illustration showing droplet deformation into a bullet-shaped droplet due to asymmetric internal pressure. (**b**) Bright-field microscopy. (**c**) ThT fluorescence image of bullet-shaped droplet. (**d**) Profile of fluorescence intensity based on fluorescence image (**c**).

**Figure 5 ijms-23-05480-f005:**
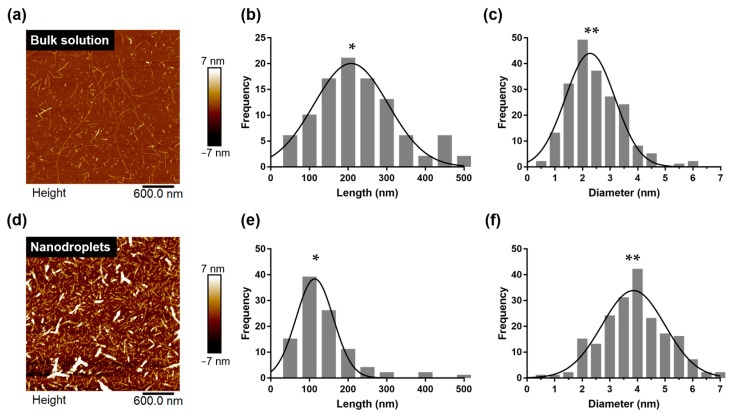
Comparison of structural information of HEWL amyloid fibrils formed in bulk solution and nanoliter droplets. (**a**) AFM image of HEWL amyloid fibrils formed in bulk solution. (**b**) Histograms of contour length. (**c**) Diameter distribution of HEWL fibrils from AFM image (**a**). (**d**) AFM image of HEWL amyloid fibrils formed in nanoliter droplets. (**e**) Histograms of contour length. (**f**) Diameter distribution of HEWL fibrils from AFM image (**d**) (* *p*-value < 0.0001; ** *p*-value < 0.0001).

## Data Availability

Not applicable.

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
