# Peer review of "Amyloid Formation in Nanoliter Droplets"

_ijms, 2022, doi:10.3390/ijms23105480_

Round 1
Reviewer 1 Report
Authors showed very interesting and original work with this study.
Only comments I have is reproducibility of some data.
-Figure 2e and Figure 3g, how many times the measurements have been done for each condition and what are the error bars and significance values?
Reviewer 2 Report
The paper is devoted to the study of amyloid formation in nanoliter droplets. The topics of the manuscript concern a very interesting problem related to microfluidic systems. The authors have proposed a simple microfluidic system to monitor the fibrillation of lysozyme from egg white. The interdisciplinary research presented in the article is sure to find many readers.
Although the article is well prepared, it is very difficult to evaluate it due to the lack of a proper conclusion.
The final section of the manuscript is clearly missing.
I recommend completing and rewriting the article.
Reviewer 3 Report
The manuscript is a description of obtained results of amyloid formation in aqueous droplets and speculations on what could be the explanations of the observed results. The authors followed the work of a different group (ref 10) who studied the formation of insulin amyloids in high aspect ratio droplets. The authors of the manuscript used different protein (lysozyme), but the same protein concentration and pH. The experimental setup allowed studies of droplets of larger volume with not as high aspect ratio. The authors observed a longer lag time of amyloid formation (about 10 times longer than the previous research by the other group), the opposite dependence of the lag time on the droplet volume, thicker and shorter amyloid fibers formed in a droplet rather than in bulk solution, and speedy formation of amyloids in the pressure-distorted parts of a droplet.
The problems this reviewer found in the manuscript are following:
- Sample preparation. The authors filtered the protein solution using a 0.2um filter before the experiment. To be sure the authors work with monodisperse protein FPLC should be used.
- Fast amyloid formation in the pressure-distorted parts of a droplet may prove that in small volumes the lag time is shorter, as was discovered previously. I do not understand why the concentration of protein should be higher in the pressure-distorted parts of a droplet. To examine the hypothesis of the effect of protein concentration on the lag time, an appropriate experiment should be performed.
- English should be corrected, there are awkward sentences in the manuscript which are difficult to understand, for example, lines 44-45, “in particular…”, or lines 48-50, “For example…”
- Since tube sealing produces a “bullet shaped” droplet, showing faster amyloid formation at the edges, the sealing procedure should be described.
- In line 80 authors say that the heat stimulated Brownian motion, but at temperature 60C and pH=2 lysozyme is unfolded, so the major effect of the temperature is protein unfolding
- Line 68, 0.05mm diameter is a typo
- Line 109, must be “cross beta sheet”
- Line 110, the “ThT intensity” is a slang
- Line 114, determination of the lag time should be discussed
- Figure 3 (e) why the fluorescence image is not circular but linear?
- Figure 3 (f) needs more data points for a reliable fit. Current data points are too sparse
- Line 129 and below. Why do authors discuss convection? I do not think there is any convection in a droplet when it is held at a temperature of 60 degrees for hours. Please check the definition of convection. The authors should prove if there is any convection in a droplet
- Line 145, Rapid amyloid formation – this needs to be confirmed, and occasional observations must not be discussed. Authors must run experiments at different concentrations to validate their conclusion
- Line 159, fluorescence microscopy is optical microscopy as well.
- In figure 5 and the text, please provide a p-value for the length and diameter of fibers, prepared in droplets and the bulk
- Line 192, I do not think that a nanoliter droplet is a confined space. This statement needs support.
Round 2
Reviewer 2 Report
The article is dedicated to the study of amyloid formation in nanoliter droplets. The subject matter of the manuscript addresses a very interesting problem related to microfluidic systems. The authors proposed a facile microfluidic system to monitor the fibrillation of hen egg-white lysozyme. The interdisciplinary research presented in this article is sure to find numerous readers.
The article is carefully crafted. The introduction provides scientific background and all necessary information about the material, and motivation seems to be fully justified. The experimental methodology is sufficient, and the results are interpreted correctly. The quality of graphics is very well.
I recommend accepting the article after a few minor corrections, as listed below:
- My suggestion is that the order of proposed sections should be as follows:
- Introduction
- Materials and Methods
- Results
- Discussion
- Conclusions
- The abbreviation ThT should be explained as thioflavin T (ThT) before the abbreviation was used alone. If the Materials and Methods section was before the Results section, the text would not need to be changed.
- I suggest to formulate a sentence with the text in bracket - (p-value between (b) and (e) < 0.0001; 191 p-value between (c) and (f) < 0.0001) in Figure 5.
Reviewer 3 Report
This reviewer is satisfied with the changes introduced into the manuscript
Author Response
Thanks to your thorough review, we were able to improve the missing parts of the figure and manuscript. We are grateful to receive your instruction.